# p21-Activated Kinase: Role in Gastrointestinal Cancer and Beyond

**DOI:** 10.3390/cancers14194736

**Published:** 2022-09-28

**Authors:** Xiaodong Li, Feng Li

**Affiliations:** Department of Cell Biology, Key Laboratory of Cell Biology, National Health Commission of the PRC, and Key Laboratory of Medical Cell Biology, Ministry of Education of the PRC, China Medical University, Shenyang 110122, China

**Keywords:** p21-activated kinase, cancer, signaling transduction, gastrointestinal cancer

## Abstract

**Simple Summary:**

Gastrointestinal tumors are the most common tumors with a high mortality rate worldwide. Numerous protein kinases have been studied in anticipation of finding viable tumor therapeutic targets, including PAK. PAK is a serine/threonine kinase that plays an important role in the malignant phenotype of tumors. The function of PAK in tumors is highlighted in cell proliferation, survival, motility, tumor cell plasticity and the tumor microenvironment, therefore providing a new possible target for clinical tumor therapy. Based on the current research works of PAK, we summarize and analyze the PAK features and signaling pathways in cells, especially the role of PAK in gastrointestinal tumors, thereby hoping to provide a theoretical basis for both the future studies of PAK and potential tumor therapeutic targets.

**Abstract:**

Gastrointestinal tumors are the most common tumors, and they are leading cause of cancer deaths worldwide, but their mechanisms are still unclear, which need to be clarified to discover therapeutic targets. p21-activating kinase (PAK), a serine/threonine kinase that is downstream of Rho GTPase, plays an important role in cellular signaling networks. According to the structural characteristics and activation mechanisms of them, PAKs are divided into two groups, both of which are involved in the biological processes that are critical to cells, including proliferation, migration, survival, transformation and metabolism. The biological functions of PAKs depend on a large number of interacting proteins and the signaling pathways they participate in. The role of PAKs in tumors is manifested in their abnormality and the consequential changes in the signaling pathways. Once they are overexpressed or overactivated, PAKs lead to tumorigenesis or a malignant phenotype, especially in tumor invasion and metastasis. Recently, the involvement of PAKs in cellular plasticity, stemness and the tumor microenvironment have attracted attention. Here, we summarize the biological characteristics and key signaling pathways of PAKs, and further analyze their mechanisms in gastrointestinal tumors and others, which will reveal new therapeutic targets and a theoretical basis for the clinical treatment of gastrointestinal cancer.

## 1. Introduction

p21-activated kinases (PAKs) are serine/threonine kinases that were originally discovered as binding proteins for small GTPases and were soon discovered to be activated as a substrate for Cdc42 and Rac1 small GTPases [1,2,3,4]. PAKs typically mediate multiple signaling pathways by binding or phosphorylating their downstream substrates, which regulates the cellular physiological processes including cell growth, cytoskeletal dynamics, cell proliferation, survival and apoptosis. Therefore, the abnormal regulation of PAKs usually leads to a variety of fatal human diseases, such as malignant tumors, heart and brain diseases, etc. The PAK family consists of six members. Although these members contain a conserved p21 binding domain (PBD) and a kinase domain, they can still be divided into two groups according to their structure and activation. Group I contains PAK1, PAK2 and PAK3, and group II includes PAK4, PAK5 and PAK6, of which, PAK5 is sometimes referred to as PAK7. The sequence conservation of group I (~80%) was higher than that of group II (~60%), and the conservation between the two groups was relatively low (Table 1). In humans, almost all members are involved in the occurrence and development of tumors, especially the representative members, PAK1 and PAK4, of the two groups that have been studied relatively extensively.

The initial reports on PAK focused on the dynamic remodeling of the cytoskeleton [8,9], and then, they quickly entered the tumor-related research stage, with a large number of PAK kinase substrates being gradually discovered [10,11]. Together, these developments have laid the foundation for exploring the significance of PAKs in many cellular processes that confer cancer phenotypes, namely cell cycle progression, invasion, metastasis, anti-apoptosis, drug resistance, angiogenesis, the epithelial–mesenchymal transition, stemness regulation and the regulation of the tumor microenvironment. PAKs are now known as key nodes in signaling pathways that originate in the cell membrane and ultimately confer cancer phenotypes on cells. This review will focus on the canonical signaling pathways of PAKs and the important role of PAKs in gastrointestinal cancer.

## 2. Structure and Activation of PAK

All members of the PAK family are structurally composed of N-terminal regulatory sequences and C-terminal kinase domains, and they are distributed with varying numbers of proline-enriched regions. The C-terminal kinase domains of all family members are conserved and relatively more conserved within groups. The detailed molecular features and differences of these are shown in Table 1. PBD and the autoinhibitory domain (AID, for group I PAKs)/AID-like pseudosubstrate sequence (PS, for Group II PAKs) are found at the N-terminus. Among them, AID of group I partially overlaps with PBD, and plays a clear and critical role in the activation of the PAKs in group I. In contrast, the PAKs of group II are not thought to contain an AID domain according to earlier studies until the research works of Ching [12] and Baskaran [13] (Figure 1). Even so, the activation process of the PAKs in the two groups are different. 

Group I PAKs are activated via the modification of an autoinhibitory dimer which is comprised of two PAK molecules in an asymmetric antiparallel manner [14]. The two PAKs bind together in reverse, with the AID of one occupying the surface of the kinase domain of the other, therefore causing the two to inhibit each other. The binding of activators, such as activated Rac and Cdc42, to the PBD domain and concomitant interactions with proximal amino acids and phosphoinositide lead to the dissociation of the AID domain from the kinase domain and disruption of the dimer, therefore promoting the distinct conformational changes of the catalytic domain [15,16]. As a result, the conformational changes ultimately lead to the exposure of activation loops that were previously hidden within the kinase domain [17]. Then, the Thr423 (for PAK1) is phosphorylated, and several additional serine sites are further phosphorylated by auto-phosphorylation to preserve the activity of the PAK. 

Unlike group I PAKs, PAK4, the representative member of group II, is continuously phosphorylated at Ser474 in the activation loop, but it is held in an inactive state by AID until the Cdc42 binding can occur [13]. Activation loop-phosphorylated PAK4 is autoinhibited by the pseudosubstrate sequence RPKP in the AID domain. The GTPase binding of PAK4 to the PBD cannot activate PAK4, but it can affect the distribution of PAK4 in cells. The binding of an additional SH3 domain to RPKP or another portions of the N-terminus releases RPKP to allow for kinase activity to occur. There are some similarities in the activation patterns of the two other members of the group II PAKs [18] (Figure 2).

## 3. PAK Signaling in Proliferation and Migration

PAK plays an important regulatory role by phosphorylating key factors in some canonical signaling pathways. As a result, the abnormal expression or activation of PAKs leads to the occurrence and development of various diseases, especially tumors. A better understanding of the pathological processes associated with the PAK signaling pathway will provide new insights into therapeutic options and drug development for the treatment and prevention of diseases.

Over the years, the important role of PAK in the proliferation and migration of cancer cells has been extensively studied. The importance of PAK is reflected in the fact that once it is activated, the PAK signaling pathway penetrates into many important signaling pathways and important physiological processes of cells. Cell proliferation and cell migration are not only two important physiological processes of cells, but also, they are the basis for the occurrence and development of tumors. In these processes, the WNT/β-catenin signaling pathway, MAPK pathway and PI3K/AKT pathway via EGFR and HER2, respectively, are well studied. In recent years, a very large number of PAK target proteins that are related to proliferation and migration signaling pathways have been exposed, therefore suggesting that PAKs play a crucial role in these processes. In this section, we only discuss two representative members of two groups of PAK: PAK1 and PAK4 (Figure 3). PAK1 directly phosphorylates the key factors in these signaling pathways, including Raf/MEK [19], AKT [20] and β-Catenin [21], which activate downstream of the transcription factors to participate in cell proliferation. Although PAK4 can only phosphorylate β-Catenin [6], it has been reported that PAK4 can activate AKT or MEK/ERK in various tumor processes. Furthermore, other transcription factors are listed in the Figure 3 that serve as PAK1 substrates, and these participate in the processes of cell proliferation through changes in the transcriptional activity [22]. In addition to the above-mentioned substrates, PAKs also directly regulate proliferation via the cell cycle control through their target proteins, including H3, MORC2, c-Raf, Aurora A, Arpc1b, Plk1, MACK and TCoB for PAK1 [23], and Ran and p57 for PAK4 [22]. Another important role of PAKs is to participate in cell migration through the dynamic regulation of the cytoskeleton. PAKs can not only participate in the regulation of the microfilament cytoskeleton through different substrates, but also regulate the stability of microtubules through different stathmin. Some of these substrates and even the phosphorylation sites are shared by PAK1 and PAK4, such as LIMK1/2 and GEF-H1. Further, some membrane receptors such as Integrin and factors that are closely related to the receptors and the cytoskeleton, including SRC-3Δ4, FAK, Paxillin and p120-Catenin are all PAK phosphorylation substrates [22].

## 4. PAK Signaling in Survival

Apoptosis is an evolutionarily conserved cell death pathway that plays a crucial role in normal development and the maintenance of homeostasis. In addition to participating in survival signaling pathways through substrates such as JNK and NIK [24,25], PAK also negatively regulates the process of apoptosis in various ways, including through the classical death receptor pathway and the mitochondrial apoptosis pathway (Figure 4). Among all the members, PAK4 is the only known member that has been reported to regulate the death complex [26]. In addition, the regulation of apoptosis by the PAK family mainly focuses on the mitochondrial signaling pathway. Bad proteins can be phosphorylated by PAK1, 2, 4, and 5, and members of these two families share different phosphorylation sites [27,28,29,30,31]. Phosphorylated Bad proteins can further control the release of cytochrome C through the Bcl-2/Bim signaling pathway, thereby affecting the activation of Caspase9. In addition to directly phosphorylating Bad proteins, PAK1/5 can also phosphorylate Bad proteins through Raf-1. Group II members PAK5 and PAK6 can also participate in apoptosis by phosphorylating mitochondrial proteins ANT2 [32] and AIF [33].

## 5. PAK Signaling in Plasticity

The process and molecular mechanisms of the epithelial–mesenchymal transition (EMT) have been extensively studied over the years and are thought to play an important role in the initiation stage of tumor invasion, during which cells lose their differentiated epithelial properties, including cell polarization and intercellular adhesion. Instead, the cells have acquired the ability to migrate and invade, which are characteristics of mesenchymal stem cells. The representative members of the PAK family, PAK1 and PAK4, can phosphorylate the canonical hallmark of EMT, transcription factors Snail [34] and Slug [35], respectively, thereby resulting in a decreased amount of E-cadherin expression, by which, cells lose their ability to adhere to neighbor cells and further gain the ability to invade through the basement membrane and vascular endothelial cells and enter the bloodstream. In addition to Snail and Slug, PAKs ultimately induce EMT by phosphorylating other target proteins, including ELF3, CtBP, SHARP, Smad2/3, PPARγ, β-Catenin, PI3K-p85, SATB1, E47 and GATA1 [36,37,38,39,40,41,42,43,44,45]. In contrast to EMT, tumor cells can reduce the activity of PAK through some unknown regulation, restore the E-Cadherin expression of tumor cells, and thus, attach tumor cells to the metastatic organs, which is known as MET. During EMT, the primary tumor cells acquire stemness through a process that is similar to dedifferentiation. In fact, the role of PAK is not limited to EMT, but it is involved in a broader process of regulating cellular plasticity. Given that cancer stem cells (CSCs) have the same characteristics as normal stem cells, including self-renewal, differentiation, and chemoresistance, their roles in the development and progression of many cancers have become the focus of oncology research. In the PAK family, all other members except for PAK2 have been found to maintain cancer cell stemness and further participate in chemoresistance through their downstream signaling pathways including the NF-κB/IL-6, Stat3 and β-Catenin pathways [46,47,48,49,50] (Figure 5). 

## 6. PAK Regulation of Tumor Microenvironment

The tumor microenvironment (TME) refers to the extracellular environment in which tumor cells are located. Within this, in addition to the well-known extracellular matrix, blood vessels, and cytokines, there are tumor-associated fibroblasts and the immune system [51]. There is always an interaction between the tumor cells and the tumor microenvironment. On the one hand, tumor cells adapt to the microenvironment by changing their own metabolism; on the other hand, tumor cells change the microenvironment by releasing signaling factors, and at the same time, they try their best to deceive the immune cells in the microenvironment, and to prevent apoptosis. Several convincing studies have highlighted the important role of PAKs in the regulation of the tumor microenvironment and in chimeric antigen receptor T-cell immunotherapy (CAR-T) (Figure 6). PD-1 (programmed death protein 1), an immunosuppressive molecule that is expressed on T cells, is a member of the immunoglobulin superfamily. PD-1 acts as an immune checkpoint to prevent autoimmunity, mainly by promoting the apoptosis of antigen-specific T cells in lymph nodes and reducing the apoptosis of regulatory T cells. Since PD-1 binds to PD-L1 or PD-L2 to initiate the programmed cell death of T cells, tumor cells expressing PD-L1 can thus obtain an immune escape [52,53]. As early as 2012, the results of a kinome analysis in demonstrated that PAK4 was found to be a downstream target of 2B4, and that it may regulate microfilaments through its downstream pathway to participate in the immune process of NK cells [54]. The inhibition of PAK4 improves the results that are obtained by the application of chimeric antigen receptor-T cell immunotherapy (CAR-T) against cancer. On the one hand, it normalizes the tumor vascular microenvironment by the re-expression of adhesion proteins (mainly claudin-14 and VCAM-1) in endothelial cells, and on the other hand, it also attenuates the inhibition of T cell invasion that is induced by the PAK4/β-Catenin signaling pathway in cancer cells [53,55]. The inhibition of PAK1 blocks the PSC-mediated immune evasion of pancreatic ductal adenocarcinoma (PDA) cells, while reducing the intrinsic and PSC-stimulated PD-L1 expression in PDA cells, which further sensitizes PDA cells to CD8+ T cells [56]. Moreover, the expression of PAK1 in macrophages control the differentiation direction of CD4+ T cells via the IFR1/NF-κΒ/IL6 pathway, which promotes the differentiation to Th17, and inhibits the differentiation to Treg [57]. However, Treg cells stabilize themselves by expressing PAK2 to prevent their transformation to Th2-like cells (GATA3^+^), thereby maintaining their role in suppressing aggressive immune cells [58]. All of these indicate that PAKs might be involved in the regulatory system of T cells. Further, PAK4 regulates the ratio of M1/M2 tumor-associated macrophages (TAM) through HSPH1 [59], and PAK1 regulates chemotaxis and crawling along the tumor microvessel through LIMK1/Cofilin in neutrophils [60].

The tenacious vitality of tumor cells is reflected in that, on the one hand, it affects the immune system in the microenvironment in various ways to avoid cell death, and on the other hand, the cells adjusts their metabolism when there is a lack of oxygen supply. Both PAK1/2 and PAK4 are involved in the cell’s metabolism, and a cell metabolic phenotype is closely related to the TAM. In leukemia cells, PAK1 knockout resulted in glycolysis, whereas, PAK2 knockout resulted in oxidative phosphorylation [61]. Although the environment of leukemia cells is somewhat different from that of solid tumor cells, its metabolic regulation is still instructive. In fact, PAK1 regulates cellular glucose metabolism by phosphorylating phosphoglucomutase (PGM) [62], in addition to this, PAK1 binds and inhibits PGAMb thereby inhibiting its involvement in glycolysis [63]. PAK2 promotes cellular aerobic glycolysis through Pyruvate kinase M2 (PKM2) [64]. PAK4 participates in the pentose phosphate pathway through G6PD [65], and PAK4 also promotes GLUT3 expression through YAP/TAZ and promotes glucose uptake by glioblastoma cells [66]. Meanwhile, angiogenesis is another guarantee for tumor cell survival, of which, PAKs are involved. The signaling pathways HIF-1α/VEGF [67] and TGF-β/MMP-9 [68], which are downstream of PAK1, Paxillin [69] and Bmk1/Erk5 [70], which are downstream of PAK2, and ICAP1 [71], which is downstream of PAK4, are all involved in this.

## 7. PAK Profile in Cancer

The numerous targets of PAKs are at the core of many important signaling pathways, and these are also important players in biological behaviors such as cell proliferation/migration, cell survival, the regulation of cell plasticity, and cell–environment interactions. Their intracellular abnormalities, including gene mutation, gene amplification or protein overexpression, are bound to cause diseases, especially tumors, most of which are malignant and are often associated with a poor patient prognosis (Table 2).

## 8. Positive Role of PAK in Gastrointestinal Cancer

The gastrointestinal and digestive organs are susceptible to cancer and are often responsible for death. Geographical differences are one of the most important reasons for differences in the incidence of gastrointestinal cancers. According to the global data in 2020, the proportions of gastric cancer, colorectal cancer (CRC) and pancreatic cancer were 5.6%, 9.8% and 2.6%, and the death cases as a result of these were 7.7%, 9.2% and 4.7%, respectively [109].

In numerous studies on PAKs, a series of PAK kinase substrates and interacting proteins and signaling pathways have been revealed, which confirms that PAK and PAK signaling is involved in the progression of gastric cancer. The regulation of PAKs on gastric cancer progression is positive. The major target proteins and signaling pathways of PAK1 are introduced to regulate gastric cancer cell proliferation (MORC2 [110], Smad2/3 [111]), migration, metastasis and invasion (ATF2/miR-132 [112], RUFY3 [113]) and cell cycle processes (CyclinD1 [114], Cyclin B [115]). However, PAK4 mainly focuses on the regulation of the migration, metastasis and invasion (via LIMK1/Cofilin/microfilament [92], SCG10/microtubule [116], DGCR6L [117], Coro1C [118]) of gastric cancer cells. CDK12 directly binds to and phosphorylates PAK2 at Thr134/Thr169 to activate the MAPK signaling pathway in gastric cancer [119]. *H. pylori* is considered to play important roles in the development of gastric tumors, and PAKs are also involved in this process. On the one hand, an *H. pylori* infection causes PAK1 to activate the JNK signaling pathway, and on the other hand, PAK1 binds and phosphorylates NIK, which further regulates NF-κB in *H. pylori*-infected epithelial cells [120,121]. Drug resistance has always been a difficult topic in the clinical treatment of malignant tumors, mainly due to the complex and intertwined signaling pathway network in tumor cells. The complex signaling network of PAKs makes them plausible to be used as important regulators of tumor resistance. PAK2 plays roles in RhoGDI2-mediated chemoresistance in gastric cancer [122]. The PAK4-mediated activation of MEK/ERK and PI3K/AKT was validated in cisplatin-resistant gastric cancer cells [123]. However, for 5-fluorouracil/oxaliplatin resistance, there is a high expression of PAK6 in the gastric cancer cells [106]. PAK activity and the PAK-related signaling pathway was downregulated by a small-molecule drug that was designed for gastric cancer cells, and among these were: curcumin [124] for PAK1 and GL-1196 [125], and LC-0882 [126], and LCH-7749944 [127] for PAK4.

CRC has the highest morbidity and mortality rate among all gastrointestinal cancers. The signaling pathways that PAKs use to regulate the proliferation of CRC cells include classical Wnt/β-Catenin, PI3K/AKT and MEK/ERK. CDK15 binds PAK4 and phosphorylates PAK4 at Ser29, which promotes the proliferation and anchorage-independent growth of CRC cells via the β-Catenin/c-Myc and MEK/ERK pathways [128]. PB-10, a thiazolo[4,5-d] pyrimidine derivative, significantly inhibits the proliferation and colony formation of CRC cells by targeting PAK4 [129]. Small-molecule thymoquinone directly binds to PAK1 thereby changing its conformation and scaffold function, which further blocks the MEK/ERK1/2 pathway in CRC [130]. KRAS is frequently mutated in CRC, and it is closely related to the occurrence, progression, drug resistance and relapse of CRC. Interestingly, the PAK1/4-induced cell proliferation in KRAS-mutated CRC does not occur not through canonical the MEK/ERK or PI3K/AKT signaling pathway, but the anti-apoptotic signaling pathway via Bcl-2 and Bcl-X(L) [131]. In addition to the above, PAK1 also promotes the proliferation of CRC cells by regulating G1/S transition through the JNK/Cyclin D1/CDK4/6 pathway [81], or promotes the survival of CRC cells by increasing the level of HIF-1α [132]. The phosphorylation of Bad proteins by PAK5 directly or through the AKT pathway inhibits camptothecin-induced apoptosis in CRC cells [133]. PAK1 promotes MMP-7 expression, a signaling pathway that is involved in DAAM2-regulated CRC cell invasion [134]. PAK5 interacts with Integrin β1/β3 to facilitate the migration/invasion of it [104], and it phosphorylates E47 to promote the EMT of CRC cells [44]. Since the 1950s, 5-fluorouracil (5-FU) has been widely used as a basic chemotherapy drug in the treatment of colorectal cancer, which was followed by the resistance of CRC patients to 5-FU. There is no doubt that PAKs play a role in colorectal cancer chemoresistance. PAK1 overcomes 5-FU-induced G1 arrest [81], while PAK6 overcomes G2/M arrest [108]. Indeed, the PAK1-induced mechanism of CRC resistance to 5-FU is partly associated to the expression of cancer stem cell markers and the CD44 subtype profile [135].

The recognized downside of pancreatic cancer is that it has a poor prognosis [109]. The mechanism of PAKs on pancreatic cancer is mainly focused on tumor growth, EMT, and immune system regulation. Overexpression and/or overactivation of PAK1 and PAK4 in pancreatic ductal adenocarcinoma (PDAC) promotes tumor growth and metastasis [136]; aMUC13 coordinates the expression and phosphorylation of PAK1 in pancreatic cancer cells, which contribute to pancreatic tumor growth and the prospects of the individual’s survival [137]. The phosphorylation of RUNX3 by PAK1 promotes pancreatic cancer progression [138]. Dependent on the AKT and ERK signaling pathways, PAK4 in pancreatic cancer cells leads to cell proliferation and survival by promoting the nuclear accumulation of NF-κB [96]. PAK1 plays a functional role in the Met receptor tyrosine kinase-induced pancreatic adenocarcinoma growth and metastasis [84]. PAK1 modulates pancreatic cancer cell transformation as well as an invasive EMT phenotype via the NF-κB/p65/fibronectin pathway [83]. PAK4 interacts with p85α, which is a subunit of PI3K, and promotes PDAC cell motility by positively stimulating the AKT activity that is downstream of HGF [139]. PAK1 was also involved in the activation, proliferation, and apoptosis of pancreatic stellate cells, and further, the in interaction of stellate cells with cancer cells [140]. The presence of a PAK4/STAT3-induced pancreatic CSC-like property [48] implicates chemical resistance [141,142]. The inhibition of PAK1/4 increases the sensitivity of PDAC to gemcitabine both in vitro and in vivo [143,144]. Further, the knockdown of PAK4 in gemcitabine-resistant pancreatic cancer cells restores the cells’ sensitivity to the drug by upregulating hENT1 expression [145]. Unexpectedly, a subsequent retrospective study reported that there was a positive correlation between PAK4 and hENT1 in clinical samples and a negative correlation between PAK4 and a poor prognosis, which brought a trace of uncertainty to the use of PAK4 as a clinical therapeutic target for pancreatic cancer [146]. The reality is that pancreatic cancer's intractability has forced researchers to turn to immunotherapy, which is more likely to hold the promise of a treatment. Increased numbers of tumor-infiltrating CD8^+^ lymphocytes are significantly independently associated with an improved disease-free survival and the patient’s overall survival [147]. In clinical PDAC cases, PAK1, activated PSC and the CD8^+^/PAK1 ratio are all negatively associated with a favorable prognosis. The study of PAK1 in a PDAC mouse model demonstrate that the inhibition of PAK1 increases the tumor infiltration of CD4^+^ and CD8^+^ T cells, while reducing the intrinsic and PSC-induced PD-L1 expression in PDAC cells, which were then sensitized to cytotoxic lymphocytes [56]. 

## 9. PAK in Other Cancers

Other cancers, led by breast cancer, including lung cancer and prostate cancer, have the highest rates of morbidity and mortality. PAKs also play an important role in them. Looking back the role of PAKs in other cancers can undoubtedly provide ideas for the research and treatment of clinical gastrointestinal tumor pathogenesis.

## 10. PAK in Breast Cancer

PAK1/4 regulates the occurrence and development of breast cancer through downstream target proteins in breast cancer cells or tissues. PAK1 phosphorylates ER-α and NRIF3 (co-activator of ER-α), thereby leading to the estrogen responsiveness of breast cancer cells [148,149]. Both PAK1 phosphorylation CaMKII and PAK4/MEK/ERK signaling promote breast tumorigenesis [150,151]. Further, PAK1-regulated tumor growth is either dependent on classical cell cycle regulation (the phosphorylation of Histone H3 [152] and the depolymerization of α/β-tubulin [153]) or anchorage-independent growth (the activation of MAPK or phosphorylated LC8 (dynein light chain) [154,155]), or it is depend on cell survival (phosphorylated FKHR (forkhead transcription factor) [156], DLC1 and BimL [157]). In addition, ivermectin inhibits the AKT/mTOR signaling pathway through the ubiquitinal degradation of PAK1, thereby promoting autophagy in breast cancer cells [158]. The phosphorylation of β-Catenin by PAK1 is required for the ErbB2-induced efficient transformation of mammary epithelial cells [159], while, PAK4 regulates the stability of p57(Kip2), thereby resulting in the proliferation of breast cancer cells [160].

PAK1/4 regulates the invasion and metastasis [161,162,163] of breast cancer cells, either by promoting the secretion of MMP-1/3 [164], or through the LIMK1/Cofilin signaling pathway [165] or via Snail-induced EMT [34]. PAK4 phosphorylation on CEBPB [166] or Integrin β5 [167] promotes the migration and invasion of breast cancer cells. Besides this, both RUNX1 and LIFR are PAK4 substrates and promote the bone metastasis of ERα-positive breast cancer [168,169]. 

Both PAK1 and PAK4 regulate the stemness of breast cancer stem cells [47,170], which leads to drug resistance. The phosphorylation of EBP1 or CtBP by PAK1 induces tamoxifen resistance in breast cancer [72,171]. PAK1 phosphorylates ER in breast cancer cells, thereby inducing a resistance to antiestrogen therapy [172]. 

Additionally, PAK1 phosphorylates Polo Like Kinase 1 (PLK1) and promotes the p65 subunit of NF-κB to upregulate tamoxifen resistance [173,174]. PAK4-driven mammosphere-forming CSC activity increases alongside the progression of breast cancer only in the ER-positive metastatic samples. PAK4 activity increases in the ER-positive models of acquired resistance to endocrine therapies [170].

## 11. PAK in Lung Cancer

The role of PAK1/4 in lung cancer is mainly reflected in its proliferation, invasion and metastasis, and clinical drug resistance. NSCLC proliferation and invasion are mediated by PAK1/ERK, PAK/AKT and PAK1/Crk signaling [175,176,177]. However, the occurrence of squamous cell carcinoma is driven by the PAK1/CREB signaling pathway [75], and the migration and invasion of it are driven by PAK1/LIMK1/cofilin [178,179]. The phosphorylation of fumarate by PAK4 counteracts its induced cell growth arrest in lung cancer [180]. Interestingly, the JAK2-induced tyrosine phosphorylation of PAK1 appears to be involved in most cancers [181].

Resistance to tyrosine kinase inhibitors (TKIs) has become a difficult problem in the clinical treatment of tumors. Although some patients derive clinical benefit from TKI therapy due to EGFR mutations, unfortunately, most patients remain TKI resistant after treatment. PAK1 confers TKI resistance to tumor cells, whether it is in EGFR-mutant or EGFR wild-type cells [182]. Indeed, the subcellular localization of p120ctn in NSCLC determines the activity of PAK1. Cytoplasmic p120ctn constitutively activates ERK in the downstream signaling of EGFR through the activation of PAK1, and promotes cancer cell resistance to TKIs [183]. In addition to this, PAK1 induces a stemness phenotype in NSCLC cells by increasing β-Catenin expression via ERK/GSK3β, thereby resulting in the resistance of NSCLC cells to cisplatin [184].

Several PAK4 inhibitors including PF-3758309 have also demonstrated that PAK4 is involved in the proliferation and migration of lung cancer cells [185]. Due to the difference in inhibition between the two groups by different PAK inhibitors, PAK family members may act synergistically to determine lung cancer cell apoptosis or survival [186].

## 12. PAK in Prostate Cancer

TGFβ1 induces apoptosis and EMT in prostate cancer cells by activating P38-MAPK and Rac1/Pak1, respectively [187]. The metastatic promoter DEPDC1B binds to Rac1 and enhances the Rac1-PAK1 pathway to induce EMT and enhance proliferation [188]. PAK4 binds and directly phosphorylates Slug at Ser158 and Ser254 which resulted in the stabilization of Slug, and further induces EMT, and the invasion and metastasis of prostate cancer cells [35]. PAK1 promotes prostate tumor growth and microinvasion by inhibiting the transforming growth factor beta expression and enhancing MMP-9 secretion [68]. PAK1 promotes metastatic prostate cancer-induced bone remodeling [189]. PAK4 promotes prostate cancer cell migration via its kinase substrates such as LIMK1 [190]. PAK4 also regulates the focal adhesion via binding and phosphorylating paxillin of prostate cancer cells [191].

PAK6 is overexpressed in prostate cancer, localizes to cell–cell adhesion, and drives epithelial colony escape based on its kinase activity [192]. PAK6 can also directly regulate prostate cancer cell metastasis through LIMK1 [193]. PAK6 phosphorylates ANT2 and promotes ANT2 protein degradation, thereby inhibiting prostate cancer cell apoptosis [32]. Interestingly, PAK6 phosphorylates AR at Ser578, and PAK6 phosphorylates Mdm2 at Thr158/Ser186 to trigger AR degradation under androgen stimulation. Thus, PAK6 inhibits prostate tumorigenesis by regulating AR homeostasis [194].

PAK4 regulates CREB and further promotes drug resistance [195]. PAK4/LIMK1 signaling mediates the chemoprotective action of CXCL12/CXCR4, and the inhibition of PAK4 leads to re-sensitization of pancreatic cancer cells to DTX-induced cellular toxicity even in the presence of CXCL12 [196]. The knockdown of PAK6 inhibits the growth of prostate cancer growth and enhances the chemosensitivity to docetaxel [197] of prostate cancer cells.

## 13. Conclusions

Gastrointestinal tumors have received increasing attention due to their high morbidity and mortality rates. Members of the PAK family have come into the limelight shortly after their discovery, with authors blaming their anomalies for irreversibly malignant events. Here, we outline extensive scientific research and comprehensively review the mechanisms of PAKs in gastrointestinal cancers. In recent years, efforts have been made to understand the details of PAKs and PAK signaling in pathological conditions, especially in tumors. As a result, PAKs are overexpressed in various cancers which are represented by gastrointestinal tumors, causing a poor prognosis. Hyperactivated PAKs lead to a more aggressive cancer cell phenotype through numerous downstream events. As a class of oncogenic kinases, PAKs are considered to be cancer hallmarks regulating the cytoskeletal dynamics, proliferation, invasion, metastasis, plasticity, metabolism, and immune evasion of cancer cells. In fact, PAKs occupy node positions in multiple critical oncogenic signaling pathways, including PI3K/AKT, Wnt/β-catenin, ERK/MEK, and TGF-β signaling, through diverse activation pathways and numerous downstream target proteins. More importantly, PAKs were found to be involved in anti-immunity in the tumor microenvironment, revealing that there is a possibility that combining PAK small-molecule inhibitors with a CAR-T procedure may bring benefits in tumor therapy. Interestingly, it has been reported that the O-GlcNAc modification of cytoplasmic proteins regulates T cell activation [198], while PAK1-4 are all glycosylated (Table 1). Although the specific function of this glycosylation is unclear, it suggests that PAK plays a more complex role in immune evasion or immune cell infiltration. Given their role in cancer progression, PAKs have emerged as promising candidates, both as diagnostic biomarkers for the early detection of cancer and as drug targets for cancer therapy. For decades, people have devoted themselves to analyzing the crystal structure of PAK in order to reveal the biological mechanism of PAK in tumors, and further develop effective and selective small-molecule inhibitors for clinical tumor therapy. Due to the high conservation of PBD and kinase domains among the PAK family members, it is very difficult to develop highly selective PAK small-molecule inhibitors. To date, small-molecule inhibitors of PAK have not overcome clinical trials, but in any case, the important role in tumors warrants that PAK is a promising candidate target for anti-malignant tumors.

## Figures and Tables

**Figure 1 cancers-14-04736-f001:**
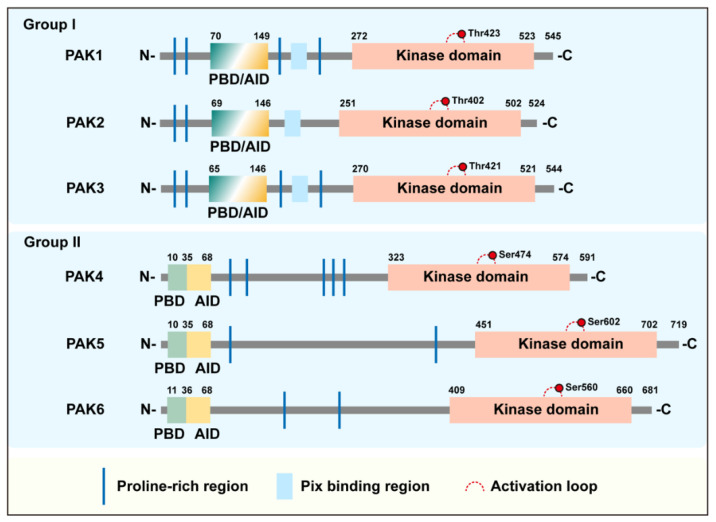
Structural organization of group I/II PAKs. All PAKs contain conserved PBD and serine/threonine kinase domains. Group I PAKs contain the classical AID, which partially overlaps the PBD, in addition to the Pix-binding region, whereas group II PAKs contain AID-like PS structures. The vast majority of PAKs contain proline-rich regions.

**Figure 2 cancers-14-04736-f002:**
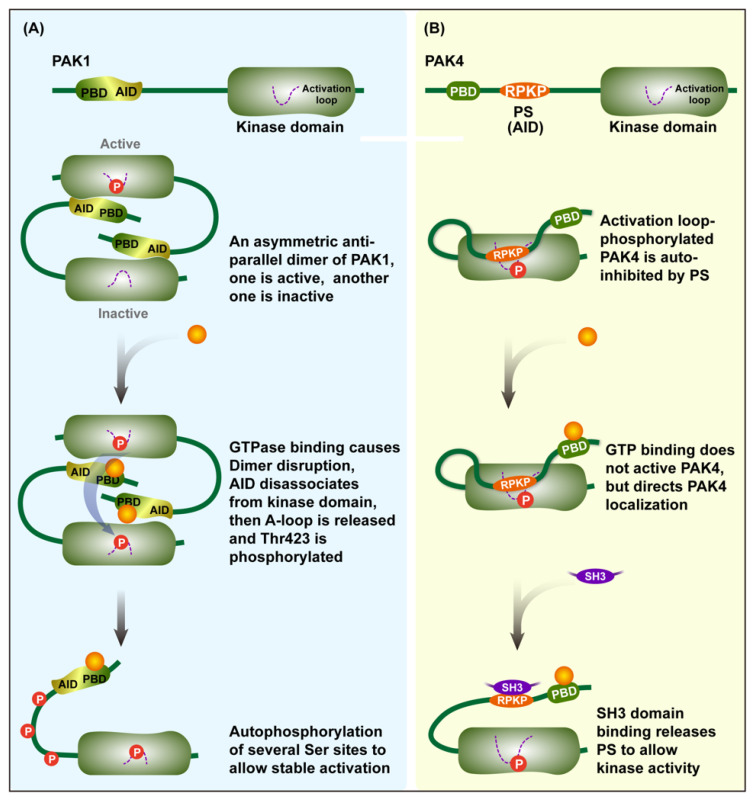
Activation model for Group I/II PAKs. (**A**) Representative PAK1 activation of group I. The AID of PAK1, which partially overlaps with the PBD, binds to the kinase domain of another PAK1 to form an antiparallel dimer. Binding of GTPase to PBD results in dissociation of AID from the kinase domain, active PAK1 phosphorylates the activation loop of inactive PAK1, and further, several Ser sites are phosphorylated to allow stable activity of PAK1. (**B**) Representative PAK4 activation of group II. PS binds to the kinase domain of PAK4 itself to inhibit the kinase activity, and further binding of GTPase does not release the activity of PAK4 but directs its localization in the cell. On this basis, the binding of SH3 domain to PS releases the kinase activity of PAK4. The red circle with the letter P indicates phosphorylating modification.

**Figure 3 cancers-14-04736-f003:**
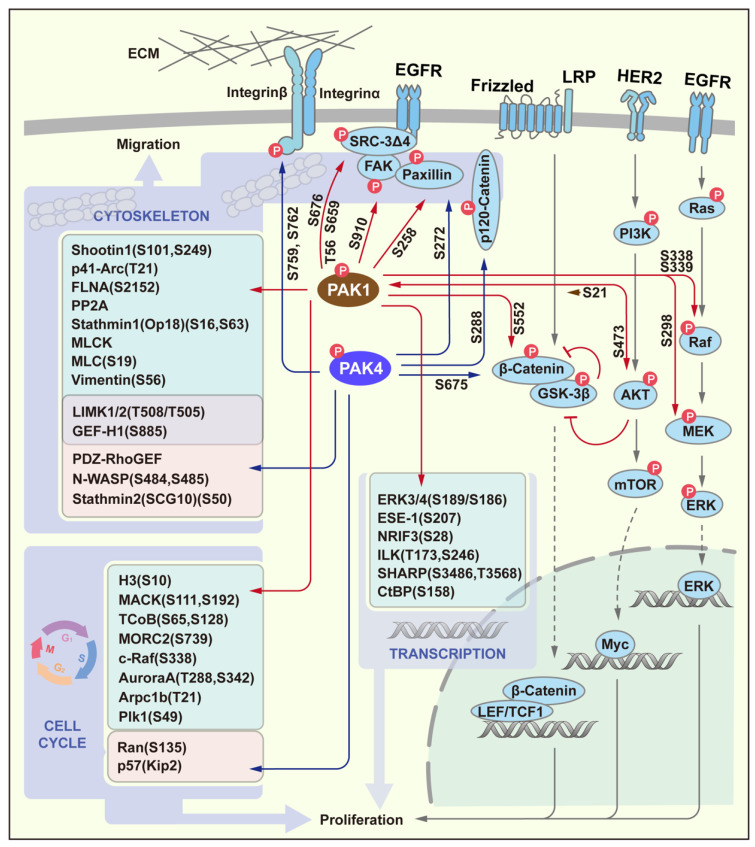
PAK signaling in proliferation and migration. PAK1 participates in the classic MEK/ERK, PI3K/AKT and Wnt/β-Catenin signaling pathways by phosphorylating key factors to promote cell proliferation. In addition, other PAK target proteins that are involved in the process of cell proliferation via the regulation of transcription or cell cycle are listed in the boxes, respectively. Substrates of PAK1/4 involved in cell migration are listed in the indicated box. The S/T numbers next to the arrowed line segments represent phosphorylation sites.

**Figure 4 cancers-14-04736-f004:**
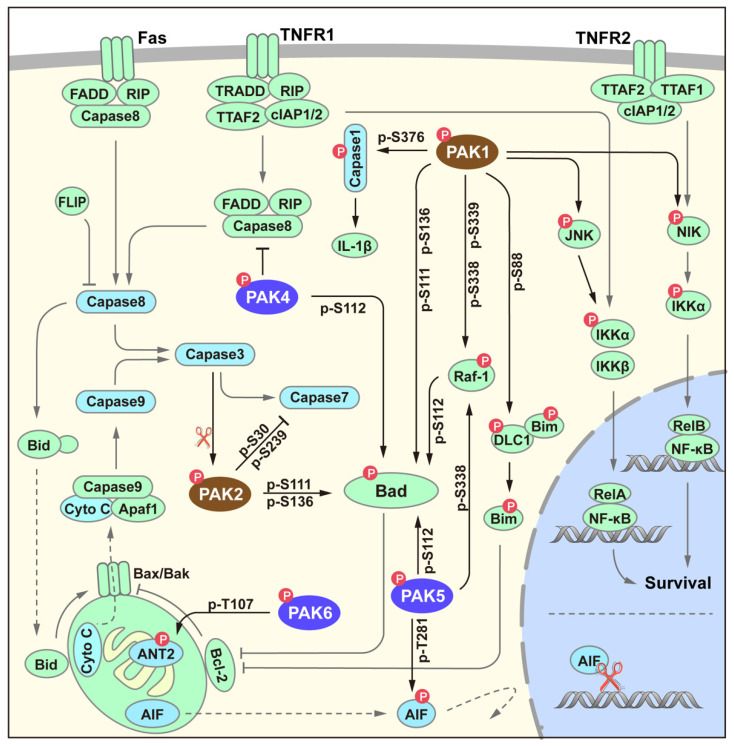
PAK signaling in survival. Grey, arrowed lines indicate canonical survival apoptotic (mitochondrial and death receptor) pathways. Black, arrowed lines indicate the regulation of PAKs in cellular anti-apoptosis and survival.

**Figure 5 cancers-14-04736-f005:**
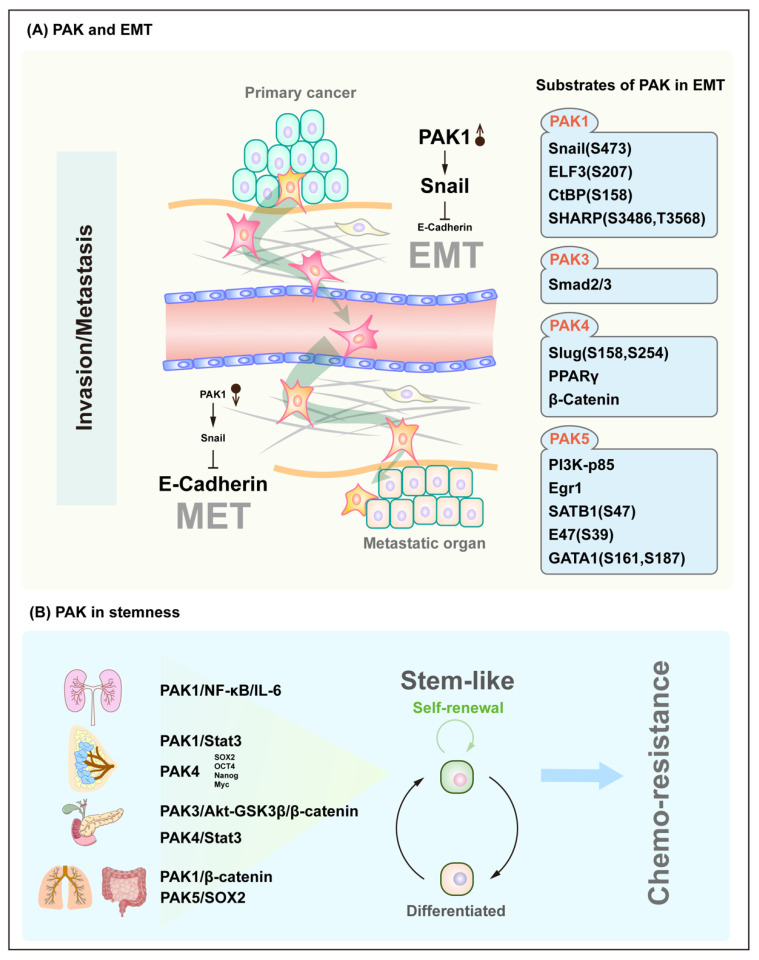
PAK signaling in plasticity. (**A**) PAK and EMT. All of the PAK downstream proteins that are involved in EMT are listed in the corresponding boxes, and each following brackets indicate the sites of PAK phosphorylation. Overexpression of PAK leads to EMT, followed by invasion and metastasis, and after reaching the metastatic organ, PAK expression is decreased, and MET occurs. (**B**) PAK and stemness. The signaling pathway of PAK in the corresponding organ leads to a stem cell phenotype.

**Figure 6 cancers-14-04736-f006:**
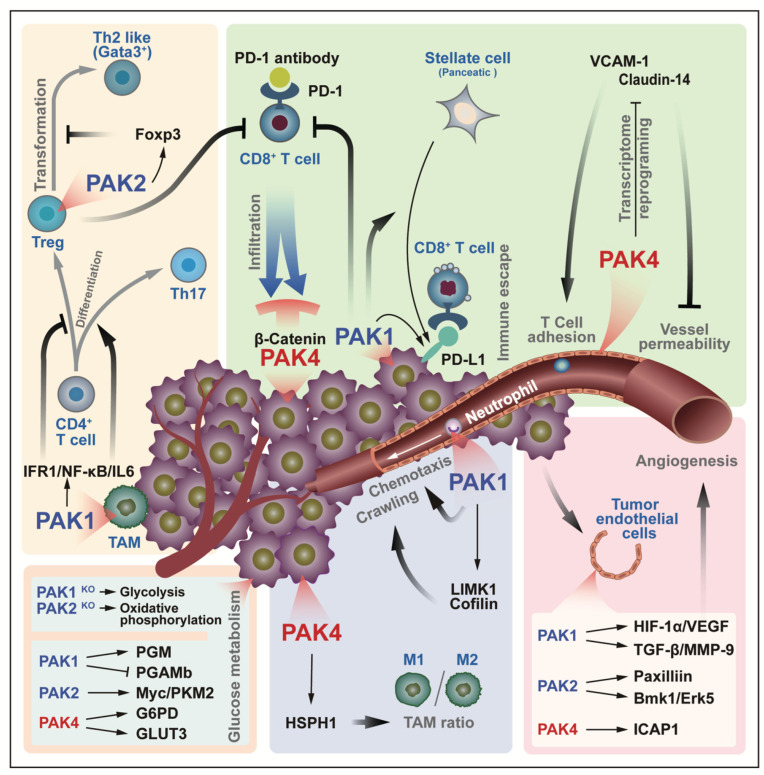
PAKs are involved in the interaction that occurs between the tumor microenvironment and tumor cells. PAK signaling prevents the immune system from attacking tumors in a variety of ways, such as promoting T cell adhesion to vascular endothelial cells, increasing vascular permeability, blocking CD8+ T cell infiltration into tumors, and enhancing immune escape through PD-1/PD-L1. In addition, the regulation of T cell differentiation, neutrophil chemotaxis, and tumor angiogenesis is also attributed to the participation of PAKs. PAKs regulate tumor cell glucose metabolism, including glycolysis and oxidative phosphorylation. All of the above processes are summed up in the boxes that have different background colors.

**Table 1 cancers-14-04736-t001:** Molecular features of PAKs.

Features		PAK1	PAK2	PAK3	PAK4	PAK5	PAK6
Similarity	To PAK1	100%	78.68%	81.87%	43.08%	41.40%	39.02%
	To PAK4				100%	62.29%	55.64%
Chemical Mass (kDa)		61	58	62	64	81	75
Crystal Structure		74–109aa 183–204aa 249–545aa	121–136aa	261–559aa	2–591aa	425–719aa	11–45aa 94–104aa 385–681aa
Localization signals	Nucleus (NLS)	243–246aa [5]	245–251aa	Not available	4–7aa, 158–161aa [6]	5–10aa [7]	Not available
	Mitochondria					30–83aa, 401–411aa [7]	
Cleavage site *			212–213aa				
Modification	Acetylation	N-Acetylation (Ser2)	N-Acetylation (Ser2)				
	Methylation				Methylation (K78)		
	Myristate		Myristate-(G213)				
	Glycosylation **	N-Glycosylation (Asn111)	O-GlcNAcylation (Thr169,178)	N-Glycosylation (Asn22,121)	O-GlcNAcylation		

Note: All data are from Uniport (PAK1-6: Q13153, Q13177, O75914, O96013, Q9P286, Q9NQU5) except for those with a reference that is marked. * PAK2 is cleavage by Caspase-3, ** data of glycosylation modification of PAKs are from Glygen (PAK1: Q13153-1,PAK2: Q13177-1PAK3: O75914-1, PAK4: O96013-1). aa—amino acid.

**Table 2 cancers-14-04736-t002:** PAK profile in human cancers.

PAK	Cancer Type	Alteration	Prognosis	Ref
PAK1	Breast cancer	Amplification Protein overexpression	decreased the relapse-free survival	[72,73]
	Lung cancer	Amplification Protein overexpression Over-activation	decreased the relapse-free survival	[74,75,76,77,78]
	Gastric cancer	Amplification Protein overexpression		[79,80]
	Prostate cancer	Protein overexpression		[68]
	Colon cancer	Protein overexpression	poor prognosis	[81,82]
	Pancreatic cancer	Protein overexpression		[83,84]
PAK2	Breast cancer	Protein overexpression		[73]
	Gastric cancer	Protein overexpression		[85]
PAK3	Lung cancer	Protein overexpression		[86]
PAK4	Breast cancer	Protein overexpression	decreased the relapse-free survival	[73,87,88,89]
	Lung cancer	Protein overexpression	decreased the relapse-free survival	[90]
	Gastric cancer	Protein overexpression Over-activation	decreased relapse-free survival	[91,92]
	Pancreatic cancer	Amplification Protein overexpression		[48,93,94,95,96]
PAK5	Breast cancer	Protein overexpression	poor prognosis	[33,97,98]
	Lung cancer	Mutation	Overall survival Immune-microenvironment	[99,100]
	Gastric cancer	Protein overexpression	poor prognosis	[101,102,103]
	Colon cancer	Protein overexpression		[104,105]
PAK6	Breast cancer	mRNA upregulation		[73]
	Gastric cancer	Protein overexpression	poor prognosis	[106]
	Prostate cancer	Protein overexpression		[107]
	Colon cancer	Protein overexpression	Decreased overall survival	[108]

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
