# Peer review of "p21-Activated Kinase: Role in Gastrointestinal Cancer and Beyond"

_cancers, 2022, doi:10.3390/cancers14194736_

Round 1
Reviewer 1 Report
Short summary should be included to the beginning of the manuscript.
Table of properties and differences of PAKs from the molecular point of view should be included into manuscript.
Legend of figures are to short.
Reviewer 2 Report
Abstract
In the abstract, the following sentences claim the review is significant and will undoubtedly inspire. I don’t think these sentences are valuable because they don’t explain the significance nor why it will undoubtedly inspire. I would remove these from the abstract. “The current research on PAKs in gastrointestinal tumors is very significant, which is attributed to the identification of a large number of downstream target proteins. Simultaneously, studies on the signaling pathways of PAKs in other cancers will undoubtedly inspire gastrointestinal cancer researchers.”
1. Introduction
Change the two “was” to “were”
p21-activated kinases (PAKs) are serine/threonine kinases that WERE first discovered as a binding protein for small GTPases and WERE soon discovered to be activated as a substrate for Cdc42 and Rac1 small GTPases
“Therefore, the abnormality of PAKs” to “Therefore, the abnormal regulation of PAKs”
Define the acronym “PBD” at its first occurrence in the paper. “Although these members contain conserved PBD domains and kinase domains”
2. Structure and activation of PAK
Change “In the initial stage, works on PAK” to “Initial reports on PAK focused on…”
Define the acronym “AID” at its first occurrence in the paper. “P21-binding domain (PBD) domain and AID (for group I PAKs)/AID”
Which group are you referring to in this sentence, Group II? “In contrast, PAKs of group are not thought to contain an AID domain by earlier studies until research works of Ching [9] and Baskaran [10].”
3. PAK signaling in proliferation and migration
The word “fully” makes it sound like this investigation is complete. “Over the years, the important role of PAK in the proliferation and migration of cancer cells has been fully studied.” I would change “fully” to “extensively” or something that doesn’t mean complete.
5. PAK signaling in plasticity
“the initiation stage of tumor invasion, during which cells lose their differentiated epithe-lial properties, including cell polarity , cell-cell adhesion” to “including cell polarization and intercellular adhesion.”
6. PAK regulation of tumor microenvironment
This long paragraph should be broken into at least two paragraphs: one for immune cells and one for metabolic changes.
“and ultimately stay away from apoptosis” to “to prevent apoptosis”
“Some convincing studies have shown that the role of PAKs in the regulation of tumor microenvironment, especially the relationship be-tween tumor and immune system, and the role of PAKs in chimeric antigen receptor T-cell immunotherapy (CAR-T) have been gradually paid more attention.” This sentence needs to be rewritten. I am not sure what the authors are trying to say here.
“through LIMK1/Cofilin of neutrophil[58].” Change “of neutrophil” to “in neutrophils”
8. PAK in gastrointestinal cancer
This statement is unclear and should be reworded: “The regulation of PAKs on gastric cancer progression is positive.”
12. PAK in prostate cancer
“TGFβ1 induces apoptosis and EMT in prostate cancer cells by activating P38-MAPK and Rac1/Pak1, respectively{Al-Azayzih, 2015 #1153}.” Reference formatting is different for several references in this paragraph.
“PAK6 can also directly regulate prostate cancer cell metastasis through LIMK1 25714010.” What is “25714010”?
Very short conclusion. “PAKs play an important role in the development of malignant tumors. Their im-portant mechanisms in physiology lead to irreversible malignant events once they are ab-normal. Whether they play a role in classical proliferation, migration, and survival, or in cell plasticity and the tumor microenvironment, which have recently received widespread attention, although small molecule inhibitors of PAK have not overcome the obstacle of clinical trials, PAKs have undoubtedly become One of the candidate inhibitory targets for defeating malignant tumors.”
